

# Global variability of carbon use efficiency in terrestrial ecosystems

**Xiaolu Tang**[1, 2, 3*], **Nuno Carvalhais**[1, 4], **Catarina Moura**[1, 4], **Bernhard Ahrens**[1], **Sujan Koirala**[1], **Shaohui Fan**[5], **Fengying Guan**[5], **Wenjie Zhang**[6, 7], **Sicong Gao**[7], **Vincenzo Magliulo**[8], **Pauline Buysse**[9], **Shibin Liu**[2], **Guo Chen**[2], **Wunian Yang**[2], **Zhen Yu**[10], **Jingjing Liang**[11], **Leilei Shi**[12], **Shengyan Pu**[3, 13], **Markus Reichstein**[1, 14, 15]

[1]Department Biogeochemical Integration, Max Planck Institute for Biogeochemistry, Jena, Germany

[2]College of Earth Science, Chengdu University of Technology, Chengdu, Sichuan, China

[3]State Environmental Protection Key Laboratory of Synergetic Control and Joint Remediation for Soil & Water Pollution, Chengdu University of Technology, Chengdu, China

[4]Departamento de Ciências e Engenharia do Ambiente, DCEA, Faculdade de Ciênciase Tecnologia, FCT, Universidade Nova de Lisboa, Caparica, Portugal

[5]Key laboratory of Bamboo and Rattan, International Centre for Bamboo and Rattan, Beijing, P.R. China

[6]State Key Laboratory of Resources and Environmental Information System, Institute of Geographic Sciences and Natural Resources Research, Beijing, China

[7]School of Life Science, University of Technology Sydney, NSW, Australia

[8]CNR - Institute for Mediterranean Agricultural and Forest Systems, Via Patacca 85, Ercolano (Napoli), Italy

[9]UMR ECOSYS, INRA-AgroParisTech, Université Paris Saclay, Thiverval-Grignon, France

[10]Department of Ecology, Evolution, and Organismal Biology, Iowa State University, Ames, IA, USA

[11]Department of Forestry and Natural Resources, Purdue University, 715 W. State St, West Lafayette, IN, USA

[12]Laboratory of Geospatial Technology for the Middle and Lower Yellow River Regions, College of Environment and Planning, Henan University, Jinming Avenue, Kaifeng, China

[13]State Key Laboratory of Geohazard Prevention and Geoenvironment Protection (Chengdu University of Technology), Chengdu, Sichuan, China

[14]German Centre for Integrative Biodiversity Research (iDiv) Halle-Jena-Leipzig, 04103 Leipzig, Germany

[15]Michael Stifel Center Jena (MSCJ) for Data-Driven & Simulation Science, 07743 Jena, Germany

**Corresponding to:** Xiaolu Tang (lxtt2010@163.com)





**Abstract**:

Vegetation carbon use efficiency (CUE) is a key measure of carbon (C) transfer from the atmosphere to terrestrial biomass, and indirectly reflects how much C is released through autotrophic respiration from the vegetation to the atmosphere. Diagnosing the variability of CUE with climate and other environmental factors is fundamental to understand its driving factors, and to further fill the current gaps in knowledge about the environmental controls on CUE. Thus, to study CUE variability and its driving factors, this study established a global database of site-year CUE based on observations from 188 field measurement sites for five ecosystem types – forest, grass, wetland, crop and tundra. The spatial pattern of CUE was predicted from global climate and soil variables using Random Forest, and compared with estimates from Dynamic Global Vegetation Models (DGVMs) from the TRENDY model ensemble. Globally, we found two prominent CUE gradients in ecosystem types and latitude, that is, CUE varied with ecosystem types, being the highest in wetlands and lowest in grassland, and CUE decreased with latitude with the lowest CUE in tropics, and the highest CUE in higher latitude regions. CUE varied greatly between data-derived CUE and TRENDY-CUE, but also among TRENDY models. Both data-derived and TRENDY-CUE challenged the constant value of 0.5 for CUE, independent of environmental controls. However, given the role of CUE in controlling the spatial and temporal variability of the terrestrial biosphere C cycle, these results emphasize the need to better understand the biotic and abiotic controls on CUE to reduce the uncertainties in prognostic land-process model simulations. Finally, this study proposed a new estimate of net primary production based on CUE and gross primary production, offering another benchmark for net primary production comparison for global carbon modelling.





## 1 Introduction

The increasing levels of atmospheric $CO_2$ concentrations and climate change have highlighted our need
to a better understanding of terrestrial carbon cycling and its responses to climate change. Gross primary
production (GPP), net primary production (NPP) and autotrophic respiration ($R_a$) are the most important
and highly related components to carbon cycling. The carbon fixed by photosynthesis is allocated to a
variety of usages in plants, including growth respiration, maintenance respiration and biomass
accumulation. The allocation proportion is highly relevant to understand ecosystem carbon stock and
carbon cycles, because it strongly affects the residence time and location of carbon in the ecosystems
(Zhang et al., 2014). For example, the carbon residence time for maintenance respiration and structural
biomass of organs varied dramatically, which could range from a few hours to decades or even centuries
(Campioli et al., 2011). Although an increasing number of researches have been conducted on carbon
exchanges in different ecosystems, unanswered questions about the fate of the carbon taken up by the
ecosystem and its relationships with the environmental variables and ecosystem types are still remained.

Carbon use efficiency (CUE), defined as the ratio of NPP to gross primary production (GPP), is an
important parameter to describe the carbon transfer from atmosphere to terrestrial biomass (Bradford and
Crowther, 2013). A CUE value of 0.5 means that 50% of acquired carbon is allocated to biomass.
Generally, NPP, which is a most direct and robust estimate, is usually calculated from the biomass
increment of wood, leaves and litter on an annual base. While GPP is very complex as it consists
photosynthetic carbon gain by all leaves, including overstory and understory, but it is typically not
measured directly (DeLucia et al., 2007). Alternatively, GPP could be calculated as the sum of NPP and
$R_a$ (DeLucia et al., 2007;Curtis et al., 2005). Therefore, due to the methodological challenging, a constant
CUE value of 0.5 has been widely used in modelling carbon cycling.

Theoretically, if $R_a$ is proportional to GPP in terrestrial ecosystems that vary in vegetation type, age,
climate and soil fertility, CUE should be constant. On the other hand, if $R_a$ is proportional to biomass,
CUE should vary with differences in allocation (DeLucia et al., 2007). However, the assumption of
constant CUE have been challenged by both field observations and modelling studies (Zhang et al.,
2009;Xiao et al., 2003), and they have found that CUE vary with ecosystem type, climate, soil nutrient
and geographic allocation (Albrizio and Steduto, 2003;Maseyk et al., 2008;Xiao et al., 2003;Zhang et al.,
2009). These variations have significant effects on landscape estimates of carbon cycling. For example,
an error of 20% of the constant CUE (0.5) used in landscape models (ranging from 0.4 to 0.6) can
misrepresent a substantial amount of carbon, comparable to the total anthropogenic $CO_2$ emissions when
scaling it to total terrestrial biosphere (DeLucia et al., 2007).

Although global distributions of GPP and NPP were established, such as MODIS and Dynamic Global
Vegetation Models (DGVMs) GPP and NPP (DeLucia et al., 2007;Zhang et al., 2009), GPP and NPP did
not change in the same pattern, leading to different changing patterns in CUE compared to GPP and NPP.
For example, a photosynthesis rate reaches its maximum at the temperature of 25-30 ºC, while the



respiration rate increases exponentially with the increase of temperature (Piao et al., 2010;Ryan et al., 1997), which results in the decrease of NPP and CUE. Using the DGVMs from the TRENDY ensemble and MODIS-derived GPP and NPP, He et al. (2018), Zhang et al. (2009) and Zhang et al. (2014) have estimated the global CUE and plotted it along a geographical and climate gradients. These studies have advanced our knowledge on understanding the global distribution of CUE, however, the validity of the conclusion might be sensitive to simplified parameters, including varying plant functional types, the constant maximum radiation conversion efficiency and respiration rate per unit of leaf and wood biomass used in different biomes and applied in these NPP and GPP product algorithms (Zhang et al., 2009).

Previous studies, based on individual observations or process-based model estimates, indicate that site fertility and management are important drivers of CUE by increasing resource availability for plants (Vicca et al., 2012;Campioli et al., 2015). However, whether these factors are dominant drivers for temporal and spatial variability of CUE has not been assessed. Additionally, atmospheric nitrogen deposition, largely overlooked before, might be another confounding factor affecting GPP (Fleischer et al., 2013) and NPP (Stevens et al., 2015;LeBauer and Treseder, 2008), and the further prediction of the spatial variability of CUE. Therefore, diagnosing the co-variation of CUE with climate and other environmental factors is fundamental to understand its driving factors, and to further fill the current gaps in knowledge about the controls on CUE.

In this study, we compiled a new dataset consisting of 415 site-year CUE observations from 188 sites distributed across all the global terrestrial ecosystems and climate regions (Fig. 1), updated from databases from Luyssaert et al. (2007) and Campioli et al. (2015), and other peer review publications prior to February 2017. For global CUE mapping and imputation, 15 global variables clarified by four types were extracted for each set of site coordinates for the measurement year (Table S1). Furthermore, we compiled additional local attributes, including climate region, site management practice, and ecosystem types. The objectives of this study were to: (1) study the ecosystem gradients of CUE; (2) explore the spatial variability of CUE and its potential climatic, edaphic, and management factors; (3) estimate CUE-derived NPP.

## 2 Materials and methods

### 2.1 Dataset

This study established a global database of site-year CUE based on observations from 188 sites extending Luyssaert et al. (2007) and Campioli et al. (2015) database. Five ecosystem types were clarified – cropland, forest, grassland, wetland and tundra. The key rule for inclusion in this database was that measurements of NPP and GPP were available for the same year, and each single year measurement was taken as an independent observation according to our selecting criteria ("*Criteria of selecting publications*" in Supporting information). NPP included both above- and below-ground growth, which could be estimated by harvest, biometric models, or increment core (below-ground). According to the



procedure of Vicca et al. (2012) and Campioli et al. (2015), gap-filling of missing NPP components, such

as understory and herb NPP, was conducted in forest ecosystems. After the gap-filling, a seven percent

increase of CUE was observed (Fig. S2).

The integrated and updated database contained 415 observations. The maximum plausible CUE was 0.84 and 8 observations were excluded after plausibility check ("*Plausibility check*" in Supporting information). Managed forest sites were excluded from modelling the temporal and spatial patterns of

CUE although there was a statistical difference on the average of CUE between managed and non-managed forests (Fig. S3), but management as a covariate contributed little to the statistical power of the RF model. Furthermore, there is scarce information on management practices globally in order to use it as an upscaling covariate, and in general the DGVMs lack also the description of the management dynamics that lead to these differences. These management sites mainly included fertilized and thinning

sites. Finally, the dataset included 286 observations for forests, 33 for grassland, 27 for wetland, 56 for cropland and 5 for tundra, which were used for one-way analysis of variance (ANOVA) to compare the significance of CUE among the five ecosystem types. Before and after removing the managed forest sites, one-way ANOVA results did not change and further indicated that CUE varied with ecosystem types (Fig. 2 and S4).

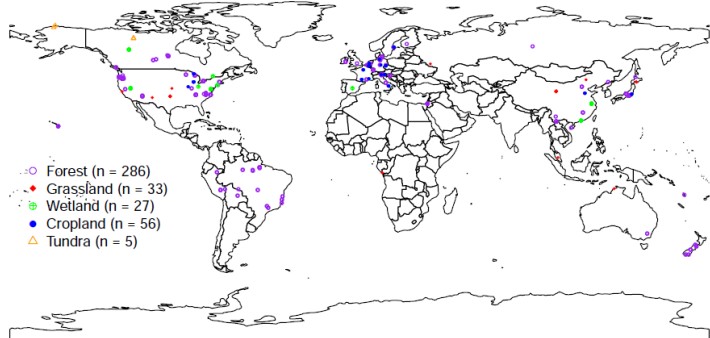


**Figure 1.** Site distribution and the number of observations for forest, grass, wetland, crop and tundra ecosystems. Geographical distribution of the observational sites in the database is not even. Western Europe has excellent coverage, while Eastern part and Russia only feature sparse sites. Asian sites are also mostly grouped on the coastal areas, while Africa is for the great part not represented. From a biome

point of view, forest sites are largely over-represented with respect to others.

### 2.2  Global variable selection

We used 15 variables of four types to predict CUE globally (Table S1). Since NPP and GPP observations were collected from the publications based on a yearly-scale, monthly climate data were





annually averaged (e.g. temperature) or summed (e.g. precipitation). Since GIMMS NDVI ranged from
July 1981 to December 2015 and LAI and fPAR ranged from July 1981 to December 2011 for GIMMS,
for the observational years that were not in these year ranges, the values of the closest year were extracted.
Soil fertility level is an important CUE driving factor (Vicca et al., 2012). However, determining the soil
fertility levels is challenging because it is determined not only by soil nutrient contents, but also by the
interaction of soil textures, pH, depth and bulk density. Therefore, in this study, soil organic matter is
used as an integrated indicator of soil fertility because it is a nutrient sink and source, enhances soil
physical and chemical properties, and promotes biological activity (de la Paz Jimenez et al., 2002).
Global land cover was taken from MOD12Q1 product (https://lpdaac.usgs.gov/data_access/data_pool).
Primarily, there were 17 land cover types. Because there were not enough observations for the each land
cover in our dataset, we aggregated the land cover into four land cover types - cropland, forest, grassland
and wetland, with a spatial resolution of 0.5º × 0.5º.

### 2.3 TRENDY Models

Since the reported of NPP and GPP in TRENDY models allowed us to test their capability of predicting
CUE, in this study, a set of 13 TRENDY models were used. These models included: CLM4.5 (Lawrence
et al., 2011), HyLand (Levy et al., 2004), ISAM (Cao, 2005), JSBACH (Kaminski et al., 2013), JULES
(Clark et al., 2011), LPJ (Sitch et al., 2003), LPJ-GUESS (Smith et al., 2001), LPX-Bern (Stocker et al.,
2013), OCN (Zaehle and Friend, 2010), ORCHIDEE (Krinner et al., 2005), TRIFFID (Cox, 2001),
VEGAS (Zeng et al., 2005) and VISIT (Kato et al., 2013).

Since most of the process models reported a monthly dynamic of NPP and GPP, to calculate CUE, the
monthly NPP and GPP were first summed to an annual scale. When comparing TRENDY-CUE of
different ecosystem types, TRENDY-CUE was aggregated to 0.5º × 0.5º for TRENDY models with
different spatial resolutions.

### 2.4 Data analysis

ANOVA analysis with a *post hoc* Tukey's HSD test was conducted to test whether CUE varied with
ecosystem type and management.

Random Forest (RF) is a machine learning approach that uses a large number of ensemble regression
trees but a random selection of predictive variables (Breiman, 2001). RF does not only consider non-
linear relationships and the interactions of the variables, but also assesses the importance value of the
variables. The importance value of a given variable is expressed by the mean decrease in accuracy (or
increase in mean square error, %IncMSE, Breiman, 2001). The higher the importance value is, the more
importance of the given variable is.

To reduce the number of the variables and improve the model efficiency, "rfcv" function within
"RandomForest" package was used in R language (Kabacoff, 2015). This function showed the cross-





validated prediction of models with sequentially reduced number of predictors (ranked by variable importance) and a 10-fold cross-validation was applied in this study. At last, six variables (ecosystem type, annual mean precipitation, annual mean temperature, nitrogen deposition, latent heat flux and diurnal temperature range, Fig. S5) were selected to predict temporal and spatial patterns of CUE using Random forest (RF). The six variables explained 49% variance in CUE.

Six types of cross-validations were conducted (Figs. S6-8): leave-one-site-out cross-validation (LOSOCV, leave all year observations within one site out and predicted by the rest site-years observations for each site), mean-site cross-validation (MSCV, building a RF model using all observations and validating mean-site CUE as a new dataset), leave-one-latitude-out (LOLOCV, leave all year observations of the same latitude out and predicted by the rest site-years observations for each latitude) and mean-latitude cross-validation (MLCV, building a RF model using all observations and validating mean-latitude CUE as a new dataset), multi-year cross-validation (MCV, validating CUE extracted from predicted CUE map only for sites with more than four-year observations with observed CUE) and "range" cross-validation (RCV, validating predicted CUE and observed CUE within the same change range of each predicting variable). These cross-validations contributed to the uncertainty of predicting the time series CUE for unknown sites. Finally, Pearson correlation efficiency, model efficiency and root mean square error were calculated.

## 3 Results and discussions

### 3.1 Ecosystem gradient of CUE

CUE varied widely from 0.201 to 0.822 (Fig. 2), while the overall mean CUE across different ecosystem types, climate regions and management practices was $0.488 \pm 0.136$ (mean $\pm$ 1 standard deviation). CUE varied significantly by ecosystem types ($p < 0.001$) with the highest value found in wetland ($0.607 \pm 0.133$), followed by tundra ($0.573 \pm 0.125$), cropland ($0.566 \pm 0.145$), forests ($0.464 \pm 0.127$) and grassland ($0.457 \pm 0.109$). Cropland and wetland CUEs were significant higher than forest and grassland, while forest CUE did not differ significant from grassland. Tundra CUE was not different from that of cropland, forest, grassland and wetland. Lower CUE in forests indicates higher respiration requirement to maintain higher forest ecosystem biomass production compared to other ecosystems (Piao et al., 2010). In comparison, the lowest CUE value was found in grassland presumably due to the heightened respiration caused by a limitation of precipitation (Shao et al., 2016). Moreover, the lack of oxygen in the saturated soil of wetland may suppress belowground Ra, while fertilization and intensive management in cropland help to increase biomass yield and reduce the respiration proportion (Campioli et al., 2015;Snyder et al., 2009). Thus, our results imply that CUE among ecosystems was not constant and a constant CUE of 0.5 could lead to biased estimates for C cycling modelling across temporal and spatial scales (see "*Practical implication for NPP estimation*" section).




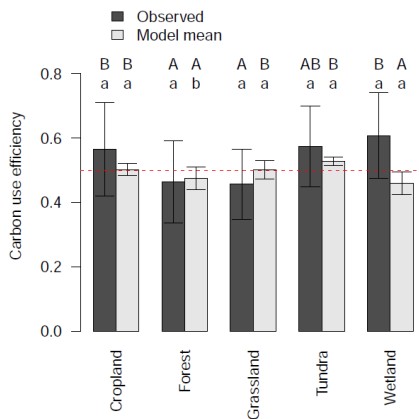

**Figure 2.** Carbon use efficiency (CUE) in cropland, forest, grassland, tundra and wetland for both observed and TRENDY CUE. The capital letters (A and B) on error bars of observed and model mean CUE indicate significant difference among five ecosystem types for observed and model mean CUE, respectively, using one-way analysis of variance (ANOVA) at $p < 0.05$; while the different lowercase letters (a and b) suggest the significance between observed and modelled CUE for each of the five ecosystem types. The red horizontal line indicates constant CUE (0.5).

However, our conclusion was different from Campioli et al. (2015), who proposed a constant CUE across ecosystems. Such different results can be attributed to: 1) different grouping strategies; and 2) a stricter filtering criteria used in our study. We grouped ecosystems in five classes (see above) due to a limited number of observations for some of individual ecosystems. In our database, we only included publications simultaneously reporting NPP and GPP in the same given year, while even at the same site, NPP and GPP reported in different years were excluded since the climatic variables can lead to significant variability in GPP (Anav et al., 2015;Jung et al., 2011) and NPP (Li et al., 2017) ("*Criteria of selecting publications*" in Supporting information). Measurements of each single year were taken as an independent observation. Additionally, a plausibility check of CUE was conducted in our database for each given year and the maximum acceptable CUE was 0.84 ("*Plausibility check*" in Supporting information).

Management practice increased CUE regardless of the ecosystem types (Fig. S3), which was consistent with Campioli et al. (2015). This is likely attributed to (1) the increase of carbon allocation to biomass production (Campioli et al., 2015); (2) the decrease of belowground C flux (Litton et al., 2007) and (3) the decrease of the allocation of GPP to Ra (Vicca et al., 2012). Regarding to the ecosystem types, management practice only increased forest CUE, rather than grass ecosystem (Fig. S3). Therefore, when modelling temporal and spatial distribution, managed forest sites were excluded (Yang et al., 2014). However, it should be noted that the one-way ANOVA results did not change when the managed forest sites were excluded (Fig. S4).




### 3.2 Spatial variability of CUE

Random Forest (RF) analysis (Breiman, 2001) indicated that ecosystem type was the most important
driving factor of CUE (Fig. S5) considering climate, satellite (GIMMS NDVI, LAI and fPAR), soil and
site variables (Table S1). This result corroborated our finding that CUE varied significantly with
ecosystem types. Across the latitudinal gradient, CUE decreased with latitude, varying from 0.58 at 65ºN
to 0.42 at 10ºS (Fig. 3). The latitudinal pattern was consistent with MODIS-based CUE, which can be
explained by the changes of temperature and CUE sensitivity to temperature (Ryan et al., 1994).
Normally, the rate of respiration increases exponentially with temperature (Ryan et al., 1994;Ryan et al.,
1995), or has a higher sensitivity to temperature compared to GPP (Curtis et al., 2005), or plants have
higher energy requirements to maintain living tissues (Ryan et al., 1994) or longer growing season with
the increasing temperature (Piao et al., 2007), while the photosynthesis rate stabilized over a wide range
of temperatures, i.e. 20–35°C (Teskey et al., 1995). Thus, plants allocate relatively more C to respiration
cost in higher temperature areas. However, the highest CUE was observed in the intensively cropped
region, such as the central North America, central Europe and North China. Nonetheless, there was no
significant latitudinal pattern of CUE for cropland ecosystem (Figs. S9), except for a few grid cells above
60ºN. This result indicated that the variations of CUE are more intensively controlled by management
practices to maximize production, hence increase CUE, while the role of climate variability was lower
in CUE variability for crops. Besides, nutrient availability was another important controlling factor of
CUE (Vicca et al., 2012). For example, tropical areas are generally constrained by soil nutrient
availability, particularly by low phosphorus concentration (Reich et al., 2009). These results further
challenged the conventional assumption that the CUE should be consistent independent of environmental
conditions (Campioli et al., 2015;Waring et al., 1998;Maier et al., 2004). However, CUE had no
significantly temporal trend during 1982-2011 (data was not shown).

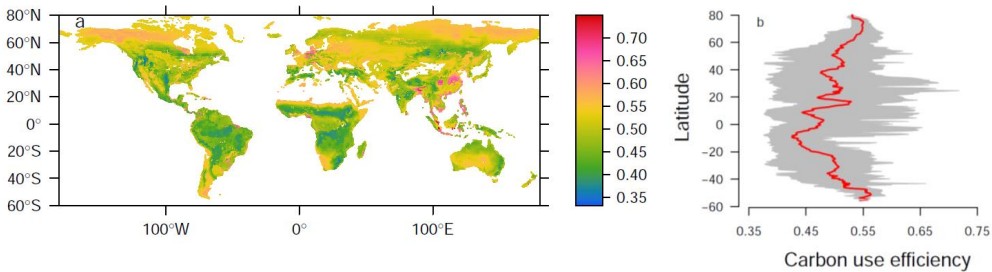

**Figure 3.** (a) Spatial distribution and (b) zonal mean carbon use efficiency (CUE) during 1982-2011
predicted by Random forest. The grey range means 2.5 to 97.5 percentile ranges of the predicted CUE.

### 3.3 Comparing with the TRENDY models



TRENDY models have been widely used to estimate the temporal and spatial variability of NPP (Shao
       et al., 2016) and GPP (Jung et al., 2017), providing a valuable tool to analyse temporal and spatial
       variability of CUE. However, due to the definition of different plant functional types among TRENDY
       models, different parameters of the same plant functional type across space were applied in different
       TRENDY models. This leads to a different magnitude and spatial patterns of GPP (Anav et al., 2015)
and NPP (Shao et al., 2016), and further affecting temporal and spatial patterns of CUE.

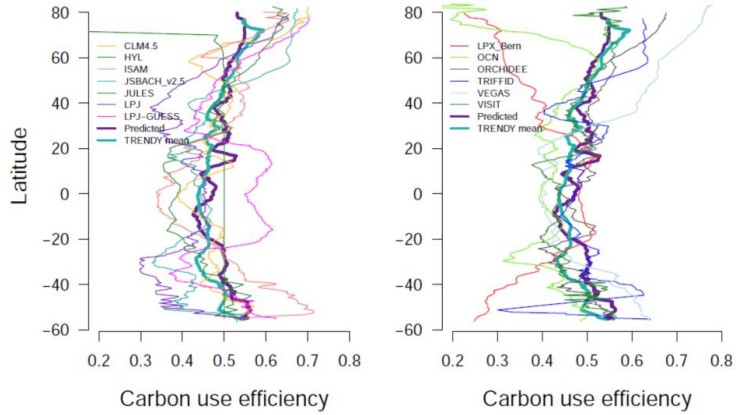

**Figure 4.** Latitudinal analysis of TRENDY carbon use efficiency (CUE). The bold purple curve
represents predicted CUE using Random forest.

       We compared CUEs derived from the 13 TRENDY model simulations for (1) the same number of
observations at the same locations sites for per ecosystem type and (2) the spatial patterns. TRENDY
       model mean CUEs varied from 0.460 for wetland to 0.527 for tundra, which had a lower change range
       compared to observations (Fig. 2). On the other hand, there was no significant difference between
       observed and TRENDY CUEs ($p = 0.0715 – 0.539$), except for forest ($p = 0.018$). However, latitudinally,
       we found a large spread among models (Fig. 4 and S10). Larger variabilities of TRENDY-CUE were
observed compared to predicted CUE and these variabilities were particularly large at high latitude
       (>60°N), suggesting that TRENDY models overestimated or underestimated CUE at high latitudinal
       areas. This result was consistent with Xia et al. (2017), which reported overestimated CUE from
       TRENDY model in permafrost areas. Eight of 13 TRENDY-CUE decreased with latitude, and OCN-and
       LPX_Bern-CUE was lowest in high latitude, while JSBACH_v2.5-, LPJ- and LPJ-GUESS-CUE showed
an increasing pattern in the topical areas. HYL-CUE was constant across all latitudes due to a fixed ratio
       (0.5) of plant respiration to total photosynthesis (Levy et al., 2004). Similar patterns were found for
       TRENDY-CUE of each ecosystem type (Fig. S11). These different CUE patterns may be related to
       several reasons:





First, different plant function types were used for different TRENDY models and constant parameter
was used for each plant function type across time and space (Xia et al., 2017).

Second, different sets of equations and parameters can lead to different estimates of GPP and NPP,
further contributing to the differences of modelled CUE (He et al., 2018). Except the HYL model, none
of the TRENDY models uses a fixed CUE, thus TRENDY-CUE was determined by the difference
between the GPP and Ra, including both maintenance and growth respiration. However, the simulated
maintenance and growth respiration varied greatly among different TRENDY models (Xia et al., 2017).
Based on a global database and upscaling tree-level Ra estimates to the stand-level, annual Ra is linearly
related to biomass (Piao et al., 2010), indicating that sites with higher biomass need higher maintenance
respiration. Although TRENDY models stimulated growth respiration dynamically, fixed growth
respiration coefficients were used, such as 0.25 for JULES (Clark et al., 2011) and TRIFFID (Cox, 2001),
0.28 for ORCHIDEE (Krinner et al., 2005) and 0.30 for CLM4.5 (Lawrence et al., 2011). Nonetheless,
using constant growth respiration coefficients in model simulation will ignore the inter-annual variability
of climatic and soil nutrient controls and generate a simplistic representation of plant respiration, which
could not describe the mechanisms of plant respiration in relation to climate change temporally and
spatially, thus causing the major source of uncertainty of CUE. For example, maintenance respiration
varies with temperature and growth respiration contributes 40-60% of total respiration in the growing
seasons (Stockfors and Linder, 1998). Even if growth or maintenance respiration acted as a constant
fraction of GPP, the respiration rate will change between years due to the variability of GPP. Therefore,
further studies are still needed to explore how maintenance and growth respiration respond to climate
change across time and space.

Third, most models do not consider nutrient constraint, such as nitrogen, which ignore the GPP or
NPP increment induced by increasing nitrogen deposition (Anav et al., 2015;Shao et al., 2016). Fourth,
due to the lack of explicit representation of $CO_2$ diffusion within leaves (Sun et al., 2014), TRENDY
models underestimate the photosynthetic responsiveness to increasing atmospheric $CO_2$ (Anav et al.,
2015). Last but not the least, since TRENDY models without representing agricultural management, crop
physiology and fertilization treatment, which are important practices to increase production (Guanter et
al., 2014), TRENDY models generally underestimated crop CUE and no model could capture the spatial
change in CUE in croplands (Fig. S11). Additionally, although the same climate data is used for all
TRENDY models to remove the uncertainty of the different meteorological forcing, using a particular
forcing can lead to systematic errors that will be propagated to the output of carbon models (Anav et al.,
2015). Therefore, our observed CUE indicated that the model predictive capability of CUE need to be
improved to better representation of the terrestrial C cycling. On the other hand, both predicted CUE and
TRENDY-CUE challenged a constant CUE and called for variable CUE for modelling global C cycling
across space and time for different ecosystem types.

### 3.4 Practical implication for NPP estimation





**Table 1.** NPP prediction from observations, MODIS and TRENDY models

| Model name | NPP (Pg C a$^{-1}$) |
|---|---|
| This study | 59.1 ± 0.2 |
| MODIS | 54.4 ± 0.9 |
| LPJ | 64.3 ± 1.6 |
| LPJ-GUESS | 54.7 ± 1.6 |
| ORCHIDEE | 47.2 ± 1.7 |
| ISAM | 52.7 ± 1.5 |
| VISIT | 55.5 ± 2.1 |
| VEGAS | 52.7 ± 1.2 |

Our results had important practical implications, particularly for estimation of global NPP. First, our study paves way to derive NPP directly from established base GPP estimates, such as Jung's GPP (Jung et al., 2017). Using this approach, global NPP estimate of this study was 59.1 ± 0.2 Pg C a$^{-1}$ (Table 1),

which is close to the reported value of 60 Pg C a$^{-1}$ of IPCC (Ciais et al., 2013). Such result highlights the potential of estimating NPP as a proportion of GPP, particularly in area of non-access and complex site structures.

Second, this study shows that using flexible CUE values improves prediction accuracy of global C cycling for different ecosystem types. Our results indicated that CUE varied spatially (Fig. 3), thus using

a constant CUE derived NPP may lead anthropogenic bias for NPP estimate. Using the modelled (spatially varied) CUE derived NPP in this study could potentially reduce the bias, therefore, such NPP estimate could serve as a 'ground truth' or benchmark. NPP estimates using Jung's GPP multiplied by constant CUE (0.5) was 61.9 ± 0.2 Pg C a$^{-1}$, which overestimated global NPP by 2.8 Pg C a$^{-1}$ (Fig. S12). This amount equals 30% of anthropogenic $CO_2$ emissions (Janssens-Maenhout et al., 2017).

Third, our NPP estimate indicates the improvement of MODIS algorithms. MODIS NPP was 54.4 ± 0.9 Pg C a$^{-1}$, which underestimated NPP by 4.7 Pg C a$^{-1}$ compared to this study, equalling 50% of anthropogenic $CO_2$ emissions (Janssens-Maenhout et al., 2017). This conclusion was also confirmed by previous study that MODIS underestimated production due to the light saturation in tropical areas (Propastin et al., 2012). Such underestimation can be also observed in Fig. S13.

Fourth, our NPP estimate highlights a better parameterization to improve the representation of processes controlling NPP in TRENDY models. We calculated TRENDY NPP as a proportion of TRENDY GPP, and NPP of different TRENDY models ranged from 47.2 ± 1.2 to 64.3 ± 1.6 Pg C a$^{-1}$ from 1982 to 2011 (Table 1). Such result indicates TRENDY models underestimated or overestimated NPP due to the simply representing growth and maintenance respiration as a proportion of GPP and

lacking of representing site management and $CO_2$ fertilization effects (Anav et al., 2015). Considering the inter-annual variability of respiration coefficient is an important step to reduce the major source of uncertainty of C flux and CUE. Last, our global CUE map facilitated ground-truthing NPP estimation,



thus providing a viable alternative of existing MODIS and TRENDY estimates of global biosphere carbon fixation rates. Our study shows that previous global MODIS NPP estimates that are based on a

fixed CUE values can be 4.7 Pg C a$^{-1}$ lower than the actual value, an underestimation that is four times greater than the total annual fossil-fuel $CO_2$ emission of the entire European Union (Janssens-Maenhout et al., 2017). Therefore, it is of great socioeconomic importance to account for the global variability of CUE in terrestrial ecosystems in estimating carbon fixation rate of the biosphere.

In summary, although data-derived CUE may serve as a benchmark for ecosystem models, directly

upscaling from observations has not been observed. This study presents an approach to fill this knowledge gap by compiling a global CUE database and predicting CUE with global environmental variables using RF algorithm, providing a global CUE product with a moderate resolution of $0.5^o \times 0.5^o$. Presently, robust findings include: (1) the pronounced CUE variation between and within different ecosystem types, challenging the perspective that CUE is independent of environmental controls; (2) a

strong spatial variability of CUE with higher CUE at higher latitudes and lower CUE in tropical areas; (3) the comparison of CUE between observed based estimates and TRENDY models, and among TRENDY models varied greatly, particularly in high latitude areas, highlighting the need for a better process representation to improve the representation of processes controlling CUE in TRENDY models. Our data analysis further indicated that the mismatch between RF-CUE and TRENDY-CUE was caused

by both (1) differences in ecosystem type (significant difference for forest ecosystem in Fig. 2); (2) differences in land cover distribution globally [e.g. different plant functional types or land overs used in TRENDY models (Xia et al., 2017)]. However, a question still remains whether such mismatch in CUE between RF and TREDNY can be also related to the misrepresentation of vegetation C stock or CUE sensitivities to environmental controls. Additionally, further improvements in the approach should

overcome shortcomings from reduced data availability and the mismatch in spatial resolution between covariates and in situ CUE.

*Code availability*: The detailed R codes are available upon the request of corresponding author.

*Data availability:* The dataset is available at: https://pan.baidu.com/s/1PxEc2a4aLHALEyWAWR0TjA or https://oc.bgc-jena.mpg.de/index.php/s/1bUKt3a2cy2Fkb0

*Author contributions.* XT, NC and MR convinced the study design. XT and CM collected data from publications. XT, NC, BA, SK, SF, FG, WZ, ZY, JL and SG contributed part of data analysis. VM and PB shared original data. XT, SL, GC, WY, LS and SP jointly proposed ideas for NPP estimate. XT prepared the manuscript with contributions from all co-authors. All authors contributed to review the manuscript.

*Competing interests*. The authors declare that they have no conflict of interest.

**Acknowledgement**





This study is supported by postdoc funding from Max-Planck-Institute for Biogeochemistry. This study is also jointed supported by the National Natural Science Foundation of China (31800365 and 41671432); the Fundamental Research Funds of International Centre for Bamboo and Rattan (1632018003 and

1632018009); Innovation funding of Remote Sensing Science and Technology of Chengdu University of Technology (KYTD201501); Starting Funding of Chengdu University of Technology (10912-2018KYQD-06910). Great thanks for all the authors' contributions of the data collection from the publications. Great thanks to Dr. Matteo Campioli for his critical comments on the dataset. "The MOD12Q1 data product was retrieved from the online Data Pool, courtesy of the NASA Land Processes

Distributed Active Archive Center (LP DAAC), USGS/Earth Resources Observation and Science (EROS) Center, Sioux Falls, South Dakota, https://lpdaac.usgs.gov/data_access/data_pool".

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
