# Peer review of "Criteria of selecting publications"

_Biogeosciences, 2019_

## Referee Comment (RC1) · Anonymous Referee #1 · 6 Apr 2019

Tang et al. investigate the global variation in carbon use efficiency (CUE), a carbon cycle property that is determined by the ratio of net to gross primary production (NPP / GPP), and thus essentially by autotrophic respiration. Based on a collection of in-situ NPP and GPP measurements and climate, vegetation and soil variables, they apply a machine learning algorithm (random forest) to derive a spatial CUE map. Subsequently, spatial gradients in this map are presented and compared to CUE simulated by dynamic global vegetation models.

Estimating vegetation CUE at the global scale is definitely urgently required to better understand and predict climate-carbon cycle feedbacks. Currently, observation based spatial estimates of CUE are lacking, mainly due to difficulties in measuring NPP even at local scales.

Unfortunately, the methods underlying the work presented here involve several important shortcomings and it is thus hardly possible to draw robust conclusions. A list of the most important issues is presented below. Apart from methodological shortcomings, also phrasing and English grammar are on a very poor level throughout the manuscript. To my mind, in its current form this study is not meeting the criteria for publication in Biogeosciences.

Major issues

1. The compiled database of in-situ NPP and GPP measurements is presented nowhere in the manuscript. Which are the additional studies other than Luyssaert et al. (2007) and Campioli et al. (2015) that are included? They should be listed in a table and properly referenced in the main text. It is thus also not evident which time span is covered by the measurements, and if temporal changes in climate or other conditions may have an effect on the compiled measurements that is neglected in this study. Moreover, is this database substantially different from or similar to the CUE data presented in Collalti and Prentice (2019, https://doi.org/10.1093/treephys/tpz034)?

2. The spatial representativeness of measurements included in this study is insufficient (see Figure 1). The complete lack of measurements in e.g. Africa and Russia can potentially lead to biases in CUE, and does not allow to robustly model global relationships between CUE, climate, vegetation, soil and management variables, since the variance and interaction of these variables cannot be covered sufficiently. It does not allow for a detailed representation of major biomes, for instance not for a division into tropical/temperate/boreal biomes. Only 5 measurements for tundra are not significant. It is also not clear how the division of the so-called "ecosystems" that are distinguished here (Forest, Grassland, Wetland, Cropland, Tundra), which are rather biomes, is implemented. For instance, how are wetlands defined?

3. It is extremely difficult to measure all NPP components in the field. NPP can hardly be estimated "directly" or in a very "robust" way (as stated in Line 69-70). What

about root NPP, herbivory, carbon allocated to reproductive parts, root exudates or VOC emissions? These can make up quite substantial percentages, depending on the ecosystem. It must be shown that NPP measurements used in this study account for all relevant NPP components and are comparable. See for instance Luyssaert et al. (2007, https://doi.org/10.1111/j.1365-2486.2007.01439.x) or Clark et al. (2001, https://doi.org/10.1890/1051-0761(2001)011[0356:MNPPIF]2.0.CO;2) for detailed discussions of this issue.

4. It is completely unclear how the measurements of CUE can be related to the independent variables used for spatial modelling. There must be an important scale mismatch that cannot be neglected (this is also indicated in the last sentence of the main text, Line 380). While NPP and GPP can only be measured for small areas (there is not even a hint on the spatial scale of measurements anywhere in the manuscript), the climate, vegetation (satellite products) and soil variables are certainly measured at a resolution of 50 km or even coarser (the spatial resolution of these variables is also not stated in Table S1). How do you account for this scale mismatch? The climate, vegetation and soil variables must be measured at the same location (and time) as the NPP and GPP measurements to allow establishing relationships and their subsequent upscaling. Moreover, why has this set of variables been selected (Table S1)? Why not including GPP or biomass estimates (according to the theory that Ra may be proportional to GPP or biomass)? Why are differences between species not taken into account? Because of the above reasons, it is not surprising that only 49% of the variance in CUE can be explained by the 6 most important variables, which is not even the majority of the variance (Line 187). The scale mismatch between the other independent variables and CUE measurements may also contribute to the finding that ecosystem type is the by far most important independent variable (Line 244, Fig. S5).

5. It is not apparent how this work leads to new insights regarding the understanding of processes shaping CUE, which would also be very relevant for a better representation of CUE in process-based models. The presented approach is designed to derive a

global CUE map based on a machine learning approach (random forest), but does not analyse processes leading to differences in observed CUE.

Specific comments

What are the main conclusions of this study? This is not evident from the abstract. E.g. Line 42: More precise main results (values of CUE) required

Line 42-43: CUE actually increases with latitude

Line 55-57: "Gross primary production (GPP), net primary production (NPP) and autotrophic respiration (Ra) are the most important and highly related components to carbon cycling." What do you mean here? Are turnover, decomposition and heterotrophic respiration less important?

Line 96-98: "Previous studies, based on individual observations or process-based model estimates, indicate that site fertility and management are important drivers of CUE by increasing resource availability for plants (Vicca et al., 2012; Campioli et al., 2015)." The studies referred to here did not use any process-based model estimates.

Line 66-68: Bradford and Crowther (2013) did certainly not invent the term "CUE". Other literature has to be cited here.

Line 69-70: I totally disagree here (see major issue 3).

Line 72-73: How is Ra measured then? I also wonder why you do not mention the possibility to measure GPP by eddy covariance and flux partitioning methods.

Line 80: What is geographic allocation?

Line 81-84: If this shall be an example calculation, what exactly is the effect of 20% error in CUE on the carbon cycle (value!)?

Line 85-86: Please explain in more detail the GPP and NPP available from MODIS or DGVMs, including more appropriate references.

Line 90: TRENDY needs to be explained (and requires a reference).

Line 109: Why are only studies until February 2017 included? 2 years have passed since then already.

Line 137-139: This is a result and does not fit in the methods section.

Fig. 2: Please double-check your ANOVA results. For instance, why is the observed CUE in Tundra grouped into AB (what does AB mean actually?), while the observed CUE in Cropland and Wetland is grouped into B despite very similar mean values. Or: Why is the observed and modelled CUE in Wetland not significantly different despite much larger differences than between observed and modelled CUE in Forest, which in turn shall be significantly different? These results seem questionable.

Line 227-234: Which selection criteria have been applied by Campioli et al. (2015)?

Line 248-249: Where do you show MODIS based CUE and how does this relate to Ryan et al. (1994)?

Line 250-256: It is not clear which temperature range is covered by the CUE measurements in your data set. In addition, another main limitation of this study is that you only account for the relation between CUE and temperature for annual mean conditions, but not the response of CUE to temperature variation during the course of the year.

Line 272: Not only different definitions of PFTs are responsible for differences in CUE simulated by TRENDY models.

Line 280: Again, how do you account for the mismatch in spatial scale between observed and modelled CUE?

Line 333: What is Jung's GPP? This needs to be introduced in detail.

Line 338-339: Where do you show this, and what exactly do you mean by "prediction accuracy of global C cycling"?

Line 339-340: "thus using a constant CUE derived NPP may lead anthropogenic bias for NPP estimate." Impossible to understand this sentence.

Line 345: "Third, our NPP estimate indicates the improvement of MODIS algorithms." How do you indicate the improvement of MODIS algorithms?

Line 350: How exactly can representations or parameterizations of processes in models be improved by your work? As far as I can see, you try to derive spatial patterns in CUE, but this work does not allow drawing conclusions on the underlying processes (see major issue 5).

Line 357: "our global CUE map facilitated ground-truthing NPP estimation". I do not understand what ground truthing and your global CUE map have in common.

Line 360: It is extremely bold to refer to your NPP estimate as "the actual value".

Line 365: As mentioned above (major issue 4), a valid upscaling of relationships between CUE and climate etc. is not achieved by this study either.

Internal server error when trying to download data from https://oc.bgc-jena.mpg.de/index.php/s/1bUKt3a2cy2Fkb0

Table S1: What does "Publications" mean? Which ones?

Fig. S12/S13: The chosen colour scale makes it hard to see the spatial differences.

---

## Referee Comment (RC2) · Anonymous Referee #2 · 10 Apr 2019

The article studies the carbon use efficiency defined as the ratio of NPP and GPP for different ecosystem types. They used a machine learning algorithm (called "Random Forest") to predict CUE from global climate and soil variables. Their results were compared with simulation output from different DGVMs. They give some explanations about difference between model output and observation and point out the importance to check for variable CUE. The article is well written and organized and fits into the scope of the journal.

General remarks:

The author used data from the TRENDY model ensemble. The differences between observed and modelled CUE is explained by model deficiencies. However new model versions are now available. Within the ISIMIP2b project there exist more up-to date

model runs. The focus in the ISIMIP project is more on future climate projections, but there have also data available for present climate. The authors should add a discussion about this. The description of the machine learning algorithms is rather brief. The algorithm should at least be described in more detail in the supplement because it is a key part of the study. The authors show in Figure 4 the latitude dependence of modelled, observed and predicted CUE. In addition some quantitative statistical measures should be shown in a table.

More specific comments:

The authors should add a mathematical definition of CUE to the main text instead of the supplement: CUE:=NPP/GPP Furthermore Figure S1 might be moved to the main text

Page 11, line 315. It is stated that most models do not consider nutrient constraints, in particular nitrogen. However, there are models with explicit nutrient limitations. There exists a version of the LPJ-GUESS model (Smith et al. 2014, Biogeosciences), e.g., that takes nitrogen limitations into account. Also the JSBACH model used in this study has an updated version with nitrogen. Perhaps it is possible to include result from these updated models into their study.

Page 11, line 311. While growth respiration is generally set to a constant in DGVMs, maintenance respiration in LPJmL, e.g., depends on air/soil temperature and C:N ratios respectively.

Page 11, line 317. Increased $CO_2$ concentration leads to better water use efficiency and therefore lower water stress increasing the productivity in DGVMs. This lead in generally to an overestimate of the $CO_2$ fertilization effect because other limitation such as nutrient limitations are not taken into account. Please comment on that.

Page 13, line 376: Typo: land over instead of land cover

Page 13, line 378: Typo: TREDNY instead of TRENDY

---

## Author Comment (AC1) · 21 May 2019

Dear Reviewer,

Thank you very much for your great efforts, comments and suggestion! According to your comments and suggestion, we revised the manuscript carefully and thoroughly. Please see, below, our point-to-point response.

Please do not hesitate to let us know if you have additional questions and/or comments.

Sincerely,

Xiaolu Tang, on behalf of all co-authors

Reviewer #1

Tang et al. investigate the global variation in carbon use efficiency (CUE), a carbon cycle property that is determined by the ratio of net to gross primary production (NPP/GPP), and thus essentially by autotrophic respiration. Based on a collection of in-situ NPP and GPP measurements and climate, vegetation and soil variables, they apply a machine learning algorithm (random forest) to derive a spatial CUE map. Subsequently, spatial gradients in this map are presented and compared to CUE simulated by dynamic global vegetation models.

Estimating vegetation CUE at the global scale is definitely urgently required to better understand and predict climate-carbon cycle feedbacks. Currently, observation based spatial estimates of CUE are lacking, mainly due to difficulties in measuring NPP even at local scales. Unfortunately, the methods underlying the work presented here involve several important shortcomings and it is thus hardly possible to draw robust conclusions. A list of the most important issues is presented below. Apart from methodological shortcomings, also phrasing and English grammar are on a very poor level throughout the manuscript. To my mind, in its current form this study is not meeting the criteria for publication in Biogeosciences.

Answer: we have addressed all the methodological issues that you pointed out in our revised manuscript (see below), and have thoroughly and carefully improved phrasing and grammar according.

Major issues

1. The compiled database of in-situ NPP and GPP measurements is presented nowhere in the manuscript. Which are the additional studies other than Luyssaert et al. (2007) and Campioli et al. (2015) that are included? They should be listed in a table and properly referenced in the main text. It is thus also not evident which time span is covered by the measurements, and if temporal changes in climate or other conditions may have an effect on the compiled measurements that is neglected in this study. Moreover, is this database substantially different from or similar to the CUE data presented in Collalti and Prentice (2019, https://doi.org/10.1093/treephys/tpz034)?

Answer: thank you for your suggestion! Additional sites with measurement years were included in the original dataset in the Figshare: https://doi.org/10.6084/m9.figshare.8157932.v1, which is publicly available.

Our data is different from Collalti and Prentice (2019) because they included CUE observations from several studies, e.g. (Campioli et al. 2015, Luyssaert et al. 2007). Besides including observations from Luyssaert et al. (2007) and Campioli et al. (2015), our study added many more observational studies directly from publications until February 2017, which provided a wider CUE and coverage. In addition, to our knowledge, we ran the first attempt of a plausibility check with the potential maximum CUE of 0.84.

2. The spatial representativeness of measurements included in this study is insufficient (see Figure 1). The complete lack of measurements in e.g. Africa and Russia can potentially lead to biases in CUE, and does not allow to robustly model global relationships between CUE, climate, vegetation, soil and management variables, since the variance and interaction of these variables cannot be covered sufficiently. It does not allow for a detailed representation of major biomes, for instance not for a division into tropical/temperate/boreal biomes. Only 5 measurements for tundra are not significant. It is also not clear how the division of the so-called "ecosystems" that are distinguished here (Forest, Grassland, Wetland, Cropland, Tundra), which are rather biomes, is implemented. For instance, how are wetlands defined?

Answer: Uneven data distribution has been a known issue in many ecological studies across the

world. We addressed this comment in the "limitations and uncertainties" section in the revised manuscript:

"Although data-derived global CUE could serve as a benchmark for global carbon cycling modelling, and this study had filled the data-gaps of data-derived CUE, limitations and uncertainties still remained in few aspects. First, uneven coverage of field CUE observations would be a source of uncertainty (Fig. 1). There was a lack of field CUE observations in Africa, Australia and Russia, but our dataset had a wide range of CUE, covering major land covers and biomes. Including more field observations in Africa, Australia and Russia would increase our capability to assess the spatial and temporal patterns of CUE in these areas. Second, there was a scale mismatch and would be a great challenge for spatial modelling, which was well justified by the site-year level predictions (Fig. S6a and c). Using a finer resolution spatial data is one possible solution to overcome this limitation. On the other hand, the study sites were globally distributed and there was a large climatic and edaphic gradient covering the major land covers and biomes, which should reflect a larger variability than the site-to-grid mismatch. Third, the gap-filling of the missing NPP component could be another error source, although this approach has been successfully applied in forest CUE in previous studies (Campioli et al., 2015; Vicca et al., 2012). However, this error source might be very small (7%, Fig. S2) according to our analysis."

We did not assess CUE for each individual biome due to limited number of observations. Then we defined five ecosystem types: Forest, Grassland, Wetland, Cropland and Tundra.

Being a wetland a distinct ecosystem that is inundated by water, either permanently or seasonally, where oxygen-free processes prevail. This information could be always observed from the publications from which we gathered information.

3. It is extremely difficult to measure all NPP components in the field. NPP can hardly be estimated "directly" or in a very "robust" way (as stated in Line 69-70). What about root NPP, herbivory, carbon allocated to reproductive parts, root exudates or VOC emissions? These can make up quite substantial percentages, depending on the ecosystem. It must be shown that NPP measurements used in this study account for all relevant NPP components and are comparable.

See for instance Luyssaert et al. (2007, https://doi.org/10.1111/j.1365-2486.2007.01439.x) or Clark et al. (2001, https://doi.org/10.1890/1051-0761(2001)011[0356:MNPPIF]2.0.CO;2) for detailed discussions of this issue.

Answer: root NPP, herbivory, carbon allocated to reproductive parts and root exudates are indeed important components of NPP. We followed approaches reported in the literature to perform a gap-filling of missing NPP components, such as understory and herb NPP, according to the procedure of Vicca et al. (2012) and Campioli et al. (2015). After the gap-filling, CUE has increased by 7% (Fig. S2). Which has been described in original manuscript.

4. It is completely unclear how the measurements of CUE can be related to the independent variables used for spatial modelling. There must be an important scale mismatch that cannot be neglected (this is also indicated in the last sentence of the main text, Line 380). While NPP and GPP can only be measured for small areas (there is not even a hint on the spatial scale of measurements anywhere in the manuscript), the climate, vegetation (satellite products) and soil variables are certainly measured at a resolution of 50 km or even coarser (the spatial resolution of these variables is also not stated in Table S1). How do you account for this scale mismatch?

Answer: to address this scale match, all the variable used in the manuscript featured the same 0.5-degree resolution. Using spatial data of a finer resolution would increase the precision of our model estimates, e.g. vegetation index. However, finer resolution spatial data are mostly not available for climate variables at global scales. Although Worldclim (reference) provides mean temperature and precipitation at 1-km resolution, the lack of annual variability and limited temporal coverage of 1970-2000 have limited its application in our study.

We address this mismatch as a separate section: "3.5 Limitations and uncertainties:

"Although data-derived global CUE could serve as a benchmark for global carbon   modelling, and this study had filled the data-gaps of data-derived CUE, limitations and uncertainties still remain. First, uneven coverage of field CUE observations is a source of uncertainty (Fig. 1). For example, there is a lack of field CUE observations in Africa, Australia and Russia. Nonetheless, the dataset encompasses a wide range of CUE, and covers major land cover types and biomes. Including more field observations in Africa, Australia and Russia would increase

our capability to assess the spatial patterns of CUE in these areas. Second, there is a mismatch of scale between observation and gridded climate and model simulations. Even though we addressed this by site-year level predictions (Fig. S6a and c), a comprehensive consideration of the effect of scale mismatch still remains a significant challenge. Using a finer resolution spatial data is one possible solution to overcome this limitation. On the other hand, the study sites were globally distributed and there was a large climatic and edaphic gradient covering the major land covers and biomes, which should reflect a larger variability than the site-to-grid mismatch".

The climate, vegetation and soil variables must be measured at the same location (and time) as the NPP and GPP measurements to allow establishing relationships and their subsequent upscaling. Moreover, why has this set of variables been selected (Table S1)? Why not including GPP or biomass estimates (according to the theory that Ra may be proportional to GPP or biomass)? Why are differences between species not taken into account? Because of the above reasons, it is not surprising that only 49% of the variance in CUE can be explained by the 6 most important variables, which is not even the majority of the variance (Line 187). The scale mismatch between the other independent variables and CUE measurements may also contribute to the finding that ecosystem type is the by far most important independent variable (Line 244, Fig. S5).

Answer: according to the definition, CUE = NPP/GPP, thus GPP and NPP are not independent of CUE. We extracted climate variables for each given year corresponding to NPP or GPP measurements. Regarding species differences, we used ecosystem types instead, given that there is no global species abundance data set and existing global models are only capable of differentiating plant functional types, or ecosystem types, but not species. On the other hand, we included GIMMS NDVI and LAI as indicators of GPP (Table S1), after a variable selection procedure, NDVI and LAI were excluded due to being less important than other variables, such as temperature.

5. It is not apparent how this work leads to new insights regarding the understanding of processes shaping CUE, which would also be very relevant for a better representation of CUE in process-based models. The presented approach is designed to derive a global CUE map based on a machine learning approach (random forest), but does not analyse processes leading to

differences in observed CUE.

Answer: although many process-model outputs have been proposed for evaluation, e.g. GPP, NPP, litter, soil heterotrophic respiration, limited effort has been placed in functional diagnostics, such as the CUE, which allows elaborating further limitations in simulating vegetation carbon stocks, or turnover times of carbon, for instance. Diagnosing different model outputs with field observations or measurements is a critical process to improve the model accuracy. Therefore, this study improves previous compilations of observation-based NPP and CUE records and associates them with climatic, soil and vegetation variables. Ultimately, these data may serve as an independent CUE to diagnose process-model-based CUE. Moreover, despite a globally robust metric (table 1), CUE shows a rather large latitudinal gradient across models (Figure 4) as the model approximates the upscaled estimates. New insights regarding processes shaping the patterns of CUE can be gained from developing and testing modeling hypotheses and evaluating them against these data.

Specific comments

What are the main conclusions of this study? This is not evident from the abstract. E.g. Line 42: More precise main results (values of CUE) required

Answer: main conclusions include: (1) CUE varied with ecosystem types; (2) CUE varied spatially with the lowest CUE in tropics, and the highest CUE in higher latitude regions; (3) both modelled CUE and TRENDY models challenged the constant CUE (0.5).

More numerical figures were added to main results:

"CUE varied with ecosystem types, being the highest in wetlands (0.607 $\pm$ 0.133) and lowest in grassland (0.457 $\pm$ 0.109) with mean CUE of 0.488 $\pm$ 0.136".

Line 42-43: CUE actually increases with latitude

Answer: done throughout the text!

Line 55-57: "Gross primary production (GPP), net primary production (NPP) and autotrophic respiration (Ra) are the most important and highly related components to carbon cycling." What do you mean here? Are turnover, decomposition and heterotrophic respiration less important?

Answer: we apologize for the improper statement. Carbon turnover, decomposition and heterotrophic respiration are also important components of carbon cycling. We revised the expression:

"Gross primary production (GPP), net primary production (NPP) and autotrophic respiration (Ra) are important and highly-related components to carbon cycling".

Line 96-98: "Previous studies, based on individual observations or process-based model estimates, indicate that site fertility and management are important drivers of CUE by increasing resource availability for plants (Vicca et al., 2012; Campioli et al., 2015)." The studies referred to here did not use any process-based model estimates.

Answer: thank you for your careful revision. We removed the words "or process-based model estimates".

Line 66-68: Bradford and Crowther (2013) did certainly not invent the term "CUE". Other literature has to be cited here.

Answer: done! DeLucia et al., 2007 was cited.

Line 69-70: I totally disagree here (see major issue 3).

Answer: we revised as follows:

"Generally, NPP is calculated as the change in ecosystem biomass over the sampling period with appropriate losses, e.g. litterfall, herbivory and other losses (Roxburgh et al., 2005)"

Line 72-73: How is Ra measured then? I also wonder why you do not mention the possibility to measure GPP by eddy covariance and flux partitioning methods.

Answer: Ra could can be measured by all respiration components (Ryan et al., 1997). The statement:

"where $R_a$ could be summed after measuring respiration components (Ryan et al., 1997). Or GPP can be measured by *eddy covariance* and flux partitioning methods (Reichstein et al., 2005)"

Line 80: What is geographic allocation?

Answer: sorry for the mistyping! We meant "geographic location".

Line 81-84: If this shall be an example calculation, what exactly is the effect of 20% error in CUE on the carbon cycle (value!)?

Answer: we thought the original argument is not strong enough, therefore, we remove this sentence.

Line 85-86: Please explain in more detail the GPP and NPP available from MODIS or DGVMs, including more appropriate references.

Answer: done.

"MODIS provides a quantitative and dynamic measurement of spatial and temporal GPP of vegetation based on a core algorithm of light use efficiency model, which requires daily inputs of incoming photosynthetically active radiation and climatic variables (Zhao and Running, 2010). On the other hand, GPP derived from DGVMs were process-based, which forced by a common set of input data sets and experimental protocol (Sitch et al., 2015)".

Line 90: TRENDY needs to be explained (and requires a reference).

Answer: we added such information in method section on "TRENDY models"!

"These models are driven by varying climate and atmospheric $CO_2$ concentration sharing a common set of input data sets and experimental protocols (Sitch et al., 2015)."

Line 109: Why are only studies until February 2017 included? 2 years have passed since then already.

Answer: after finishing the data analysis, we developed a draft text in September 2017, which was then further advanced by interactions and ample discussions with co-authors. Finally, we submitted early in 2019 after collecting all co-authors' comments and suggestion. However, our dataset until February 2017 covered a wide range of land covers and major biomes.

Line 137-139: This is a result and does not fit in the methods section.

Answer: this part removed to results section and restructured accordingly.

Fig. 2: Please double-check your ANOVA results. For instance, why is the observed CUE in Tundra grouped into AB (what does AB mean actually?), while the observed CUE in Cropland and Wetland is grouped into B despite very similar mean values. Or: Why is the observed and modelled CUE in Wetland not significantly different despite much larger differences than between observed and modelled CUE in Forest, which in turn shall be significantly different? These results seem questionable.

Answer: we double-checked ANOVA analysis and confirmed that the reported results were correct (see results below analyzed in R).

Results of observed CUE

```
> umg.aov<-aov(CUE~EcoT2,data=umg.dat)
> summary(umg.aov)
             Df Sum Sq Mean Sq F value   Pr(>F)
EcoT2         4  1.243 0.31068   21.02 3.47e-15 ***
Residuals   283  4.183 0.01478
* * *
Signif. codes:  0 '***' 0.001 '**' 0.01 '*' 0.05 '.' 0.1 ' ' 1
> |
```

```
> umg.tuk <- glht(umg.aov, linfct=mcp(EcoT2="Tukey"))
> summary(umg.tuk)

         Simultaneous Tests for General Linear Hypotheses

Multiple Comparisons of Means: Tukey Contrasts

Fit: aov(formula = CUE ~ EcoT2, data = umg.dat)

Linear Hypotheses:
                        Estimate Std. Error t value Pr(>|t|)
Forest - Cropland == 0  -0.134030   0.018775  -7.139   <0.001 ***
Grassland - Cropland == 0 -0.118455   0.026682  -4.440   <0.001 ***
Tundra - Cropland == 0  -0.002104   0.056749  -0.037    1.000
Wetland - Cropland == 0  0.031838   0.028486   1.118    0.777
Grassland - Forest == 0  0.015575   0.023162   0.672    0.957
Tundra - Forest == 0     0.131926   0.055182   2.391    0.107
Wetland - Forest == 0    0.165868   0.025219   6.577   <0.001 ***
Tundra - Grassland == 0  0.116351   0.058348   1.994    0.247
Wetland - Grassland == 0 0.150293   0.031551   4.764   <0.001 ***
Wetland - Tundra == 0    0.033942   0.059195   0.573    0.976
* * *
Signif. codes:  0 '***' 0.001 '**' 0.01 '*' 0.05 '.' 0.1 ' ' 1
(Adjusted p values reported -- single-step method)
```

ANOVA analysis of observed and modelled CUE

```
> aov.forest <- aov(CUE~Model.mean,data = subset(new.dat,EcoT =="Forest"))
> summary(aov.forest)   #### p = 0.0177
             Df Sum Sq Mean Sq F value Pr(>F)
Model.mean    1  0.090 0.09022   5.692 0.0177 *
Residuals   284  4.501 0.01585
* * *
Signif. codes:  0 '***' 0.001 '**' 0.01 '*' 0.05 '.' 0.1 ' ' 1
>
> aov.cropland <- aov(CUE~Model.mean,data = subset(new.dat,EcoT =="Cropland"))
> summary(aov.cropland)   ###  p = 0.183
             Df Sum Sq Mean Sq F value Pr(>F)
Model.mean    1 0.0379 0.03790   1.823  0.183
Residuals   54 1.1228 0.02079
>
> aov.grassland <- aov(CUE~Model.mean,data = subset(new.dat,EcoT =="Grassland"))
> summary(aov.grassland)   ###  p = 0.0715
             Df Sum Sq Mean Sq F value Pr(>F)
Model.mean    1 0.0384 0.03844   3.483 0.0715 .
Residuals   31 0.3421 0.01104
* * *
Signif. codes:  0 '***' 0.001 '**' 0.01 '*' 0.05 '.' 0.1 ' ' 1
>
> aov.tundra <- aov(CUE~Model.mean,data = subset(new.dat,EcoT =="Tundra"))
> summary(aov.tundra)   ###  p = 0.539
             Df  Sum Sq  Mean Sq F value Pr(>F)
Model.mean    1 0.00860 0.008595   0.479  0.539
Residuals    3 0.05385 0.017949
>
> aov.wetland <- aov(CUE~Model.mean,data = subset(new.dat,EcoT =="Wetland"))
> summary(aov.wetland)
             Df Sum Sq  Mean Sq F value Pr(>F)
Model.mean    1 0.0087 0.008682   0.481  0.494
Residuals   25 0.4513 0.018051
```

The capital letters (A and B) on error bars of observed and modelled mean CUE indicate

significant differences among five ecosystem types for observed and TRENDY model mean CUE, respectively, using one-way analysis of variance (ANOVA) at $p < 0.05$; while the different lowercase letters (a and b) stand for the significant difference between observed and modelled CUE for each of the five ecosystem types. The red horizontal line indicates constant CUE (0.5).

Line 227-234: Which selection criteria have been applied by Campioli et al. (2015)?

Answer: (1) GPP from eddy covariance or processed models were included from Campioli et al. (2015); (2) NPP was obtained from harvest or biometric methods and process-based models; (3) multi-year measurements within one site were averaged. While in our study, we only included GPP and NPP values from measurements and did not include these values from process models; multi-year measurement of GPP or NPP were taken as independent observations. See supplementary "**Criteria of selecting publications**".

Line 248-249: Where do you show MODIS based CUE and how does this relate to Ryan et al. (1994)?

Answer: sorry, a missing reference was added (Zhang et al., 2014).

"The latitudinal pattern was consistent with MODIS-based CUE (Zhang et al., 2014), which can be explained by the changes of temperature and CUE sensitivity to temperature (Ryan et al., 1994)."

Line 250-256: It is not clear which temperature range is covered by the CUE measurements in your data set. In addition, another main limitation of this study is that you only account for the relation between CUE and temperature for annual mean conditions, but not the response of CUE to temperature variation during the course of the year.

Answer: MAT ranged from -13 to 28 °C in our dataset; however, MAT during summer would be higher than 28 °C. Although our dataset did not account for temperature variation during the year, we included diurnal temperature range as a climatic driver to model CUE (Table S1), which could reflect the temperature variation during the course of the year to some degree.

Line 272: Not only different definitions of PFTs are responsible for differences in CUE

simulated by TRENDY models.

Answer: yes, different algorithms and parameterizations would be other important differences in CUE.

"However, due to the differences in the definitions of different plant functional types, process representations and parameterizations used in TRENDY models"

Line 280: Again, how do you account for the mismatch in spatial scale between observed and modelled CUE?

Answer: as we mentioned above, we have to admit that the mismatch is a potential limitation for spatial modelling of CUE.

Line 333: What is Jung's GPP? This needs to be introduced in detail.

Answer: Jung's GPP is an upscaling model product from eddy covariance towers with a half degree resolution using Model Tree Ensemble. Therefore, we used a new term – MTE GPP. We modified the text as follows:

"such as Model Tree Ensemble (MTE) GPP (an upscaling GPP product from eddy covariance towers with a half degree resolution, Jung et al., 2017)".

Line 338-339: Where do you show this, and what exactly do you mean by "prediction accuracy of global C cycling"?

Answer: sorry to the mistake! We simply meant NPP.

"Second, this study shows that using variable CUE improves prediction accuracy of NPP for different ecosystem types".

Line 339-340: "thus using a constant CUE derived NPP may lead anthropogenic bias for NPP estimate." Impossible to understand this sentence.

Answer: thank you for careful revision. We meant a bias of NPP estimate.

"The results indicated that CUE varied spatially (Fig. 3), which may lead to a bias in NPP if a

constant CUE is used."

Line 345: "Third, our NPP estimate indicates the improvement of MODIS algorithms." How do you indicate the improvement of MODIS algorithms?

Answer: We removed this section.

Line 350: How exactly can representations or parameterizations of processes in models be improved by your work? As far as I can see, you try to derive spatial patterns in CUE, but this work does not allow drawing conclusions on the underlying processes (see major issue 5).

Answer: although many outputs of most process-models have been proposed, e.g. GPP, NPP, litter, soil heterotrophic respiration, and most process-models validated the model output by only GPP or NPP, most of other outputs were not validated. Diagnosing different model outputs with field observations or measurements is a critical process to improve the model accuracy. Actually, when selecting the drivers for modelling CUE, only variables having physical correlation to CUE were selected, and vice versa. Therefore, these selected variables could reveal the underlying process of CUE. For example, we found a decreasing trend of CUE with the increasing temperature globally. Thereby, this study developed an observation-based modelled CUE with climatic, soil and vegetation variables, which could serve as an independent CUE to diagnose process-model based CUE, and lead to new insights regarding the understanding of CUE.

Line 357: "our global CUE map facilitated ground-truthing NPP estimation". I do not understand what ground truthing and your global CUE map have in common.

Answer: we removed "ground-truthing", and rephrased as follows:

"Last, our global CUE map facilitated NPP estimation, thus providing a viable alternative to existing MODIS and TRENDY estimates of global biosphere carbon fixation rates".

Line 360: It is extremely bold to refer to your NPP estimate as "the actual value".

Answer: thank you for your careful revision. We used "observation value" instead.

Line 365: As mentioned above (major issue 4), a valid upscaling of relationships between CUE

and climate etc. is not achieved by this study either.

Answer: as we answered in major issue 4, the mismatch of the footprint and environmental variables is a big challenge for spatial modelling, not only in this study, but many other global modellings, e.g. (Hashimoto et al., 2015; Jung et al., 2011; Tramontana et al., 2016). Using a finer resolution spatial data is one possible solution, e.g. vegetation index. However, finer resolution spatial data, e.g. 10 km, is not available for climate variables at global scales. On the other hand, the study sites were globally distributed and there was a large climatic and edaphic gradient covering the major land covers and biomes, which should reflect a larger variability than the site-to-grid mismatch.

Internal server error when trying to download data from https://oc.bgcjena.mpg.de/index.php/s/1bUKt3a2cy2Fkb0

Answer: sorry! Since I have already left Max-Planck-Institute for Biogeochemistry, the cloud folder was no longer accessible. The replaced the broken link with the following link (https://doi.org/10.6084/m9.figshare.8157932.v1) works fine.

Table S1: What does "Publications" mean? Which ones?

Answer: we meant literature reports from which we collected CUE values. See Table S1.

Fig. S12/S13: The chosen colour scale makes it hard to see the spatial differences.

Answer: the colors were changed.

**References**

Campioli, M., Vicca, S., Luyssaert, S., Bilcke, J., Ceschia, E., Chapin Iii, F. S., Ciais, P., Fernández-Martínez, M., Malhi, Y., Obersteiner, M., Olefeldt, D., Papale, D., Piao, S. L., Peñuelas, J., Sullivan, P. F., Wang, X., Zenone, T., and Janssens, I. A.: Biomass production efficiency controlled by management in temperate and boreal ecosystems, Nat. Geosci., 8, 843-846, http://dx.doi.org/10.1038/ngeo2553, 2015.

Hashimoto, S., Carvalhais, N., Ito, A., Migliavacca, M., Nishina, K., and Reichstein, M.: Global spatiotemporal distribution of soil respiration modeled using a global database, Biogeosciences, 12, 4121–4132, http://dx.doi.org/10.5194/bgd-12-4331-2015, 2015.

Jung, M., Reichstein, M., Margolis, H. A., Cescatti, A., Richardson, A. D., Arain, M. A., Arneth, A., Bernhofer, C., Bonal, D., Chen, J. Q., Gianelle, D., Gobron, N., Kiely, G., Kutsch, W., Lasslop, G., Law, B. E., Lindroth, A., Merbold, L., Montagnani, L., Moors, E. J., Papale, D., Sottocornola, M., Vaccari, F.,

and Williams, C.: Global patterns of land-atmosphere fluxes of carbon dioxide, latent heat, and sensible heat derived from eddy covariance, satellite, and meteorological observations, J. Geophys. Res. Biogeosci., 116, G00J07, http://dx.doi.org/10.1029/2010jg001566, 2011.

Jung, M., Reichstein, M., Schwalm, C. R., Huntingford, C., Sitch, S., Ahlstrom, A., Arneth, A., Camps-Valls, G., Ciais, P., Friedlingstein, P., Gans, F., Ichii, K., Jain, A. K., Kato, E., Papale, D., Poulter, B., Raduly, B., Rodenbeck, C., Tramontana, G., Viovy, N., Wang, Y. P., Weber, U., Zaehle, S., and Zeng, N.: Compensatory water effects link yearly global land $CO_2$ sink changes to temperature, Nature, 541, 516-520, http://dx.doi.org/10.1038/nature20780, 2017.

Reichstein, M., Falge, E., Baldocchi, D., Papale, D., Aubinet, M., Berbigier, P., Bernhofer, C., Buchmann, N., Gilmanov, T., and Granier, A.: On the separation of net ecosystem exchange into assimilation and ecosystem respiration: review and improved algorithm, Glob. Chang. Biol., 11, 1424-1439, 2005.

Roxburgh, S. H., Berry, S. L., Buckley, T. N., Barnes, B., and Roderick, M. L.: What is NPP? Inconsistent accounting of respiratory fluxes in the definition of net primary production, Funct. Ecol., 19, 378-382, 10.1111/j.1365-2435.2005.00983.x, 2005.

Ryan, M. G., Linder, S., Vose, J. M., and Hubbard, R. M.: Dark respiration of pines, Ecological Bulletins, 43, 50-63, 1994.

Ryan, M. G., Lavigne, M. B., and Gower, S. T.: Annual carbon cost of autotrophic respiration in boreal forest ecosystems in relation to species and climate, J. Geophys. Res. Atmos., 102, 28871-28883, http://dx.doi.org/10.1029/97JD01236, 1997.

Sitch, S., Friedlingstein, P., Gruber, N., Jones, S. D., Murray-Tortarolo, G., Ahlström, A., Doney, S. C., Graven, H., Heinze, C., Huntingford, C., Levis, S., Levy, P. E., Lomas, M., Poulter, B., Viovy, N., Zaehle, S., Zeng, N., Arneth, A., Bonan, G., Bopp, L., Canadell, J. G., Chevallier, F., Ciais, P., Ellis, R., Gloor, M., Peylin, P., Piao, S. L., Le Quéré, C., Smith, B., Zhu, Z., and Myneni, R.: Recent trends and drivers of regional sources and sinks of carbon dioxide, Biogeosciences, 12, 653-679, 10.5194/bg-12-653-2015, 2015.

Tramontana, G., Jung, M., Schwalm, C. R., Ichii, K., Camps-Valls, G., Ráduly, B., Reichstein, M., Arain, M. A., Cescatti, A., Kiely, G., Merbold, L., Serrano-Ortiz, P., Sickert, S., Wolf, S., and Papale, D.: Predicting carbon dioxide and energy fluxes across global FLUXNET sites with regression algorithms, Biogeosciences, 13, 4291-4313, 10.5194/bg-13-4291-2016, 2016.

Vicca, S., Luyssaert, S., Penuelas, J., Campioli, M., Chapin, F. S., 3rd, Ciais, P., Heinemeyer, A., Hogberg, P., Kutsch, W. L., Law, B. E., Malhi, Y., Papale, D., Piao, S. L., Reichstein, M., Schulze, E. D., and Janssens, I. A.: Fertile forests produce biomass more efficiently, Ecol. Lett., 15, 520-526, http://dx.doi.org/10.1111/j.1461-0248.2012.01775.x, 2012.

Zhang, Y., Yu, G., Yang, J., Wimberly, M. C., Zhang, X., Tao, J., Jiang, Y., and Zhu, J.: Climate-driven global changes in carbon use efficiency, Global Ecol. Biogeogr., 23, 144-155, http://dx.doi.org/10.1111/geb.12086, 2014.

Zhao, M., and Running, S. W.: Drought-induced reduction in global terrestrial net primary production from 2000 through 2009, Science, 329, 940-943, 2010.

---

## Author Comment (AC2) · 21 May 2019

Dear Reviewer,

Thank you very much for your great efforts, comments and suggestion! According to your comments and suggestion, we revised the manuscript carefully and thoroughly. Please see, below, our point-to-point response.

Please do not hesitate to let us know if you have additional questions and/or comments.

Sincerely,

Xiaolu Tang, on behalf of all co-authors

Referee #2

The article studies the carbon use efficiency defined as the ratio of NPP and GPP for different ecosystem types. They used a machine learning algorithm (called "Random Forest") to predict CUE from global climate and soil variables. Their results were compared with simulation output from different DGVMs. They give some explanations about difference between model output and observation and point out the importance to check for variable CUE. The article is well written and organized and fits into the scope of the journal.

General remarks:

The author used data from the TRENDY model ensemble. The differences between observed and modelled CUE is explained by model deficiencies. However new model versions are now available. Within the ISIMIP2b project there exist more up-to date model runs. The focus in the ISIMIP project is more on future climate projections, but there have also data available for present climate. The authors should add a discussion about this.

Answer: thank you for the good suggestion! We used more recent TRENDY models (v3) in the revised manuscript. Since we could not assess a significant temporal global mean CUE, we did not consider a future projection of CUE, and we would rather focus on the spatial pattern of CUE.

The description of the machine learning algorithms is rather brief. The algorithm should at least be described in more detail in the supplement because it is a key part of the study.

Answer: more descriptions about RF were added in the main text in the method section:

"RF is an ensemble learning method which constructs a multitude of decision trees at training time and outputting the mean predicted values for the response variable. RF is fully data-driven, and does not require initial assumptions on data distribution and independency. RF does not only consider non-linear relationships and the interactions of the variables, but also assesses the importance value of the variables. In this study, we calibrated two hyper-parameters, namely the number of variables sampled as candidates for each split, and the number of trees. Moreover, RF regression can deal with a large number of features and help feature selection based on importance values (Jian et al., 2018)".

The authors show in Figure 4 the latitude dependence of modelled, observed and predicted CUE. In addition some quantitative statistical measures should be shown in a table.

Answer: We added a correlation figure between predicted and TRENDY – CUE in Fig. 4.

[Figure]

**Figure 4.** Latitudinal analysis (a, b) of TRENDY carbon use efficiency (CUE), and (c) the correlation coefficients between predicted CUE and TRENDY CUE. The numbers mean correlation coefficient.

More specific comments:

The authors should add a mathematical definition of CUE to the main text instead of the supplement: CUE =NPP/GPP. Furthermore Figure S1 might be moved to the main text

Answer: "CUE = NPP/GPP" was added.

"Carbon use efficiency (CUE), defined as the ratio of NPP to gross primary production (GPP, CUE = NPP/GPP)"

We keep Figure S1 in the supplementary since Figure S1 is closely related to the plausibility check and this placement contribute to make this concept easier to understand to readers.

Page 11, line 315. It is stated that most models do not consider nutrient constraints, in particular nitrogen. However, there are models with explicit nutrient limitations. There exists a version of the LPJ-GUESS model (Smith et al. 2014, Biogeosciences), e.g., that takes nitrogen limitations into account. Also the JSBACH model used in this study has an updated version with nitrogen. Perhaps it is possible to include result from these updated models into their study.

Answer: thank you. We used a TRENDY model v3, since we could not access the most updated version. According to the model developers, then models were included in the TRENDY v3, including CLM4.5, CABLE, ISAM, JULES, LPJ, LPJ-GUESS, LPX-Bern, ORCHIDEE, VEGAS and VISIT.

I contacted model developer, LPJ-GUESS in TRENDY v3 did not include nitrogen limitation in the model, but LPJ-GUESS included nitrogen limitation in later versions.

Page 11, line 311. While growth respiration is generally set to a constant in DGVMs, maintenance respiration in LPJmL, e.g., depends on air/soil temperature and C:N ratios respectively.

Answer: thank you. LPJmL was not in TRENDY v3. We also remove the sentence for easy understanding. Page 11, line 317. Increased $CO_2$ concentration leads to better water use efficiency and therefore lower water stress increasing the productivity in DGVMs. This lead in generally to an overestimate of the $CO_2$ fertilization effect because other limitation such as nutrient limitations are not taken into account. Please comment on that.

Answer: thank you for the good comments. However, whether an overestimate or underestimate

productivity depends on the relative change of GPP due to effects of $CO_2$ fertilization or lower water stress on GPP, which may vary with ecosystem types or biomes. Additionally, normally, temperature increases with increasing $CO_2$, which leads to the increase of autotrophic respiration via maintenance respiration (Rm). If the relative change in GPP is larger than Rm, it could be the $CO_2$ effects and warming could play compensating roles.

Page 13, line 376: Typo: land over instead of land cover

Answer: done!

Page 13, line 378: Typo: TREDNY instead of TRENDY

Answer: done!

**Reference**

Jian, J., Steele, M. K., Thomas, R. Q., Day, S. D., and Hodges, S. C.: Constraining estimates of global soil respiration by quantifying sources of variability, Glob. Chang. Biol., 24, 4143-4159, http://dx.doi.org/10.1111/gcb.14301, 2018.